# Heparin-Based Growth Factor Delivery Platforms: A Review

**DOI:** 10.3390/pharmaceutics17091145

**Published:** 2025-09-01

**Authors:** Ji-Feng Wang, Jeng-Shiung Jan, Jin-Jia Hu

**Affiliations:** 1Department of Mechanical Engineering, National Yang Ming Chiao Tung University, Hsinchu 30010, Taiwan; www.806640420.en11@nycu.edu.tw; 2Department of Chemical Engineering, National Cheng Kung University, Tainan 70101, Taiwan; 3Hierarchical Green-Energy Materials (Hi-GEM) Research Center, National Cheng Kung University, Tainan 70101, Taiwan

**Keywords:** heparin, growth factor delivery, nanocarriers, stimuli-responsive systems, tissue engineering, microspheres, nanofibers, hydrogels, regenerative medicine, drug release platforms

## Abstract

Heparin-based delivery platforms have gained increasing attention in regenerative medicine due to their exceptional affinity for growth factors and versatility in structural and functional design. This review first introduces the molecular biosynthesis and physicochemical diversity of heparin, which underpin its binding selectivity and degradability. It then categorizes the delivery platforms into microspheres, nanofibers, and hydrogels, with detailed discussions on their fabrication techniques, biofunctional integration of heparin, and release kinetics. Special focus is given to stimuli-responsive systems—including pH-, enzyme-, redox-, thermal-, and ultrasound-sensitive designs—which allow spatiotemporal control over growth factor release. The platform applications are organized by tissue types, encompassing soft tissue regeneration, bone and cartilage repair, neuroregeneration, cardiovascular regeneration, wound healing, anti-fibrotic therapies, and cancer microenvironment modulation. Each section provides recent case studies demonstrating how heparin enhances the bioactivity, localization, and therapeutic efficacy of pro-regenerative or anti-pathologic growth factors. Collectively, these insights highlight heparin’s dual role as both a carrier and modulator, positioning it as a pivotal component in next-generation, precision-targeted delivery systems.

## 1. Introduction

Growth factors (GFs) are endogenous signaling proteins that play indispensable roles in fundamental biological processes such as embryogenesis, tissue homeostasis, wound healing, and regeneration [1]. Acting through receptor-mediated signaling cascades, GFs such as vascular endothelial growth factor (VEGF), fibroblast growth factor (FGF), bone morphogenetic proteins (BMPs), and transforming growth factor-β (TGF-β) regulate a range of critical cellular and physiological processes, including angiogenesis [2,3], cell proliferation [4,5], extracellular matrix (ECM) remodeling [6,7], and immune modulation [8,9]. Owing to their profound biological potency, GFs are widely valued as key therapeutic agents in regenerative medicine. Their applications are particularly significant in fields like orthopedics [10,11], cardiovascular repair [12,13], neuroregeneration [14,15], and chronic wound healing [16,17].

Despite the therapeutic promise of GFs, their clinical translation remains limited due to several critical barriers. A primary issue is their extremely short half-lives in vivo, often lasting only minutes to hours, which renders them susceptible to rapid enzymatic degradation [18]. Another major concern is their nonspecific biodistribution. When GFs spread throughout the body rather than targeting the specific area, it can lead to unwanted side effects, including off-target effects, uncontrolled cellular signaling, or even pro-tumorigenic responses when overdosed [19,20]. Furthermore, achieving therapeutic benefits often requires frequent or high-dose administrations, which drives up costs and raises safety concerns for patients [21]. Compounding these issues, GFs are also very sensitive to formulation and environmental stress, which makes it challenging to retain their essential bioactivity during storage and delivery [22].

These limitations highlight a need for advanced delivery platforms that can achieve three main goals: stabilize GFs within the environment of the body, control their spatial–temporal release, and maintain their bioactivity throughout the entire delivery process [23]. Over the past two decades, various structurally defined delivery systems have been developed to meet these requirements, including hydrogels [24], microspheres [25], and electrospun fibers [26]. Hydrogels form hydrated 3D networks that are well suited for injectable, cell-encapsulating, and minimally invasive applications [27]. Microspheres provide a large surface-area-to-volume ratio and precise control over loading and release, making them ideal for sustained or pulsatile delivery [28]. Nanofibers mimic the ECM to support cell adhesion and localized GF presentation, especially in regenerative interfaces [29]. Each of these platforms offers distinct advantages for protecting GFs and presenting them in a biologically relevant manner.

Heparin, a highly sulfated glycosaminoglycan in the ECM, carries a greater negative charge density than other glycosaminoglycans such as chondroitin sulfate or hyaluronic acid [30]. This high charge density underlies its strong affinity for many heparin-binding proteins, including VEGF, FGF, and BMPs, via electrostatic interactions with their positively charged domains [31]. Such binding not only stabilizes GFs and protects them from degradation but also allows heparin itself to be incorporated into polymer-based delivery systems through various strategies, thereby facilitating more controlled and sustained GF release in vivo [30,32]. Heparin can be integrated into these systems through four primary strategies. Electrostatic adsorption involves the reversible binding of GFs to heparin via charge interactions, allowing for sustained release while maintaining bioactivity [33,34]. Physical blending mixes heparin directly into a polymer matrix [35]; covalent grafting chemically attaches heparin to a carrier surface [36,37]; and network crosslinking incorporates heparin into a 3D polymer network via enzymatic, chemical, or ionic bonding strategies [38]. These modification modes allow precise tuning of GF retention, release kinetics, and spatial presentation [30,39], while also enhancing receptor engagement and prolonging therapeutic efficacy [40]. In addition, heparin itself confers intrinsic anti-inflammatory, anticoagulant, and pro-angiogenic properties, making it a multifunctional component central to the rational design of regenerative delivery platforms [41].

In this review, we provide a comprehensive summary of heparin-based delivery platforms for GF therapeutics. Emphasis is placed on three key areas: (Section 2) strategies for heparin modification and functionalization—detailing how heparin is chemically or physically altered to enhance its performance; (Section 3) classification of platforms based on structural organization—examining formats such as hydrogels, fibers, and particles; and (Section 4) function-driven design principles—highlighting how these platforms are engineered to meet specific biological requirements, including sustained release, the coordinated delivery of multiple factors, and responsiveness to various stimuli. By integrating recent advancements in material design and biological performance, we aim to establish a logical framework to guide the development of next-generation GF delivery systems with translational potential.

## 2. Heparin Composition and Modification Principles

Heparin is a linear, unbranched polysaccharide composed of repeating disaccharide units, each consisting of a hexuronic acid—either L-iduronic acid (IdoA) or D-glucuronic acid (GlcA), which are C-5 epimers—and D-glucosamine (GlcN). Figure 1 illustrates a generalized disaccharide unit of heparin, highlighting key modifiable positions. Functional substitutions may occur at the 2-O-position and 6-O-position of the sugar residues (R_1_ and R_2_), the C2 amino group of glucosamine (R_3_), and the C6 hydroxyl (R_4_), allowing for variable patterns of O- and N-sulfation or N-acetylation. This structural flexibility underlies heparin’s chemical heterogeneity and its capacity to interact with diverse biological targets. These structural modifications give rise to a highly anionic polysaccharide with substantial chemical heterogeneity. These anionic groups are primarily responsible for electrostatic interactions with cationic proteins and also participate in ionic crosslinking with multivalent ions or polycations in material systems. The polymer backbone contains several chemically active groups, each enabling specific modification strategies as summarized in Table 1.

In particular, many GFs, such as VEGF and FGF, are positively charged at physiological pH and contain heparin-binding domains that enable electrostatic complexation with the negatively charged sulfate and carboxylate groups on heparin. These interactions not only stabilize the conformation of GFs but also protect them from proteolytic degradation and promote receptor-mediated signaling [42]. The binding affinity between heparin and various GFs can vary widely depending on sulfation patterns and molecular context, with dissociation constants (K_D_) typically ranging from nanomolar to low micromolar. For instance, VEGF-A exhibits a K_D_ of approximately 11–80 nM when binding to heparin [43], while FGF-2 and FGF-18 show affinities of around 130 nM and 38 nM, respectively [44]. These electrostatic associations are essential for regulating GF bioavailability and form the mechanistic basis for many heparin-based delivery systems [45].

These functional groups not only define heparin’s binding selectivity and biological activity but also determine the chemical reactivity available for material integration [41]. Building on this chemical versatility, heparin can be incorporated into delivery systems through multiple interaction modes [46]. The choice of strategy not only depends on the available functional groups but also on the desired balance between structural stability, bioactivity preservation, and release control. We categorize heparin modification strategies into four major types based on the nature of their interaction with the carrier material: electrostatic adsorption, physical blending, covalent grafting, and network crosslinking, as shown in Table 2.

### 2.1. Electrostatic Adsorption

Electrostatic adsorption represents one of the most straightforward and widely adopted strategies for incorporating heparin into nanodelivery systems [59,60,61]. This method leverages the intrinsic negative charge of heparin, attributed to its abundant sulfate and carboxylate groups, to facilitate interactions with positively charged surfaces or biomolecules. For example, Han et al. used heparin-based multilayer nanofilms to improve bFGF encapsulation, showing enhanced retention with trilayer designs [62]. Cho et al. applied in situ electrostatic binding to immobilize heparin–bFGF on polymer films, maintaining bioactivity for stem cell support [60]. In parallel, self-assembly approaches have gained attention as an electrostatically driven method to construct polyelectrolyte nanoparticles with heparin. For instance, Wang et al. constructed lipid-enclosed coacervate systems formed via electrostatic assembly between PEAD and heparin, which stabilized the structure and enabled dual release of VEGF-C and FGF-2 while preserving protein bioactivity and colloidal stability [33,34]. García-Briones et al. reported the formation of polyelectrolyte nanoparticles by complexing a synthetic diblock copolymer, poly(HPMA-b-APMA), with heparin through self-assembly [47]. These non-covalent strategies maintain the native bioactivity of heparin while avoiding harsh chemical modifications, and their reversible nature allows for environment-responsive release, particularly suitable for short-term delivery or injectable, in situ-forming systems. However, the inherent weakness of electrostatic interactions renders them susceptible to competitive displacement by serum proteins or shifts in ionic strength in vivo. Without additional crosslinking or structural reinforcement, such systems may exhibit premature or burst release, undermining their long-term efficacy.

### 2.2. Physical Blending

Physical blending refers to the incorporation of heparin into delivery systems without the formation of covalent bonds. In this approach, heparin is mixed with other polymers or matrix-forming materials, such as gelatin [63], alginate [48], hyaluronic acid [50], or synthetic polymers [49,64], during the fabrication process. This strategy allows for the homogeneous dispersion of heparin throughout the material bulk, enabling bioactive modulation without requiring chemical modification. For instance, Zhao et al. embedded heparin into poly(aldehyde guluronate) hydrogels to achieve sustained bFGF release and promote vascularization [48], while Safikhani et al. fabricated 3D-printed alginate/hyaluronic acid/gelatin scaffolds loaded with heparin to enhance endothelial cell growth [49]. Lei et al. further demonstrated that blending heparin into methacrylated hyaluronic acid microgels allowed efficient binding and slow release of PDGF-BB and TGF-β3, enhancing stem cell recruitment and chondrogenesis [50]. The main advantages of physical blending include simplicity, preservation of biomolecule integrity, and avoidance of potentially harsh chemical reactions. However, due to the absence of covalent anchoring, there remains a risk of heparin leaching under physiological flow conditions, resulting in uncontrolled GF release and reduced bioavailability.

### 2.3. Surface Modification

Surface modification refers to the chemical functionalization of polymeric carriers, scaffolds, or nanoparticles to enhance interfacial bioactivity while preserving the core material structure. Among various strategies, covalent grafting is one of the most widely adopted approaches, involving the stable conjugation of heparin via covalent bond formation [65]. This is particularly advantageous for applications requiring long-term heparin retention, such as sustained GF binding, anticoagulant activity, or anti-inflammatory response. Surface modification through covalent grafting has been widely applied to the outer layers of nanoparticles [53], fibers [66,67], hydrogels [68], and bulk materials [69,70], enabling site-specific heparin immobilization without altering internal structure. In practice, covalent surface functionalization has been implemented across a range of material systems. For example, Caracciolo et al. immobilized heparin on the luminal surface of PLLA/polyurethane (PU) vascular grafts via urethane-based coupling, which enhanced endothelialization while inhibiting smooth muscle cell proliferation [71]. Wu et al. used mussel-inspired dopamine chemistry to immobilize heparin on Fe_3_O_4_ magnetic nanoparticles, achieving sustained bFGF release and accelerated wound healing through M2 macrophage polarization [51]. Click chemistry has also enabled effective heparin and VEGF conjugation on aligned nanofiber-modified PLLA scaffolds, promoting endothelial proliferation and reducing platelet adhesion [51,52]. Song et al. reported that heparin/chitosan nanoparticles covalently grafted to cardiovascular stents improved endothelial cell growth and reduced thrombogenicity [53]. Additionally, Sui et al. utilized Ar–H_2_O–NH_3_ plasma technology to activate PTFE surfaces, facilitating subsequent covalent immobilization of heparin via EDC/NHS coupling [72]. While surface modification via covalent grafting provides high stability and customizability, it requires careful control of reaction parameters to preserve the biological activity of heparin. Factors such as substitution degree, linker length, and steric accessibility influence the availability of active sulfate domains for GF binding.

### 2.4. Network Crosslinking Involving Heparin

Network crosslinking strategies involving heparin aim to form stable three-dimensional polymer networks in which heparin plays a structural or functional role [31]. Unlike simple blending or surface modification, crosslinking yields interconnected matrices that enhance mechanical integrity, control degradation, and regulate release kinetics [73]. Depending on the crosslinking mechanism, these systems can be categorized into enzymatic, chemical, ionic, or physical crosslinking. Enzymatic crosslinking [74,75,76,77] offers high specificity under mild aqueous conditions, preserving bioactivity and allowing cell encapsulation. It enables site-selective network formation through catalytic bonding between phenol or amine groups but may be limited by enzyme stability or diffusion constraints. Chemical crosslinking [54,55,56] forms covalent bonds with excellent stability and tunability, supporting long-term applications and sustained release. However, it requires careful control of reaction conditions to avoid excessive modification that may impair biological function. Ionic crosslinking [57,58] leverages electrostatic interactions between heparin’s anionic groups and multivalent cations. It is advantageous for rapid, reversible gelation and in situ delivery systems, though its mechanical strength and long-term integrity are generally lower than covalent methods. Physical crosslinking [30,78,79] relies on non-covalent forces such as hydrogen bonding or electrostatic assembly. It is mild and reagent-free, making it ideal for sensitive biomolecules, but it offers limited structural durability and often requires supplementary stabilization.

### 2.5. Multiple Mechanisms Involving Heparin

While individual strategies—such as electrostatic adsorption, covalent grafting, or network crosslinking—offer distinct advantages, recent advances have increasingly emphasized hybrid platforms that integrate multiple mechanisms to synergistically enhance delivery performance [80,81]. These multi-mechanism systems combine the reversibility and bioactivity preservation of non-covalent interactions [82] with the stability and specificity of covalent or crosslinked architectures [83]. For example, electrostatically assembled nanoparticles may be further embedded within chemically crosslinked hydrogels [84], or physically blended systems may incorporate localized covalent grafting to prevent premature leaching. Such hierarchical designs enable staged or dual-phase release profiles, strengthen mechanical properties, and offer programmable responsiveness to environmental cues. This integrative approach provides superior control over spatial distribution, release kinetics, and biological function, making it particularly valuable for complex regenerative or therapeutic applications where multiple signaling phases are required [85].

## 3. Heparin Delivery Platforms

Given heparin’s unique ability to bind and stabilize a wide range of GFs through electrostatic interactions, it has become a central component in designing nano-scale delivery systems aimed at enhancing therapeutic efficacy and tissue specificity. Among the numerous carrier systems available, microspheres, nanofibers, and hydrogels stand out as three major classes of platforms that complement each other in terms of structure, function, and clinical applicability [27,28], as shown in Figure 2. These systems differ not only in morphology and drug loading strategies, but also in their release kinetics, tissue integration, and responsiveness to external stimuli. Microspheres offer high surface-area-to-volume ratios and excellent control over encapsulation and release, making them suitable for sustained or pulsatile delivery [28]. Nanofibers, by mimicking ECM architecture, facilitate cell guidance and local biofactor presentation, particularly in regenerative tissue interfaces [29]. Hydrogels, on the other hand, provide 3D hydrophilic networks ideal for cell encapsulation, injectable formulations, and minimally invasive therapies [27]. Each platform also allows for the integration of heparin via different physical or chemical strategies, thereby enabling tailored release profiles and biofunctionality. To comprehensively explore heparin’s role in delivery, this chapter systematically reviews these three major platform types—heparin-loaded microspheres, nanofibers, and hydrogels—highlighting their fabrication strategies, structure–function relationships, and biomedical applications in GF delivery.

### 3.1. Heparin-Loaded Microspheres and Fabrication Strategies

Microspheres are spherical, micron- to submicron-sized particulate systems composed of natural or synthetic polymers [89]. Owing to their tunable size, high surface-area-to-volume ratio, and ability to encapsulate both hydrophilic and hydrophobic agents, microspheres have become a widely utilized platform in drug delivery and regenerative medicine [90,91]. When loaded with heparin, these systems provide not only anticoagulant functionality and controlled release of bioactive molecules [92] but also enhanced GF retention and improved biological interactions at the cellular interface [93].

A broad array of fabrication techniques has been developed to prepare heparin-loaded microspheres, each offering distinct benefits in terms of particle size control, encapsulation efficiency, scalability, and material compatibility [94]. This chapter discusses six categories of fabrication strategies, beginning with five well-established methods: emulsion–solvent evaporation/extraction, spray drying, microfluidics, coacervation, and electrospraying, as shown in Table 3. These approaches differ in mechanism, process complexity, and the physical structure of the resulting microspheres.

#### 3.1.1. Emulsion–Solvent Evaporation/Extraction

The emulsion–solvent evaporation or extraction method is one of the most widely used techniques for fabricating polymeric microspheres, particularly for encapsulating hydrophilic agents like heparin [106]. In this process, a polymer-containing organic phase is emulsified with an aqueous phase, and then the organic solvent is removed by evaporation or extraction into a nonsolvent medium [107]. Depending on the solubility of the encapsulated drug, either a single (oil-in-water, O/W) or double (water-in-oil-in-water, W/O/W) emulsion can be employed—hydrophobic drugs often use single emulsions, whereas water-soluble compounds (like heparin) require W/O/W double emulsions to provide an inner aqueous compartment. In the context of heparin, which is highly water-soluble, W/O/W double emulsion systems are typically used for encapsulation [108]. In a typical W/O/W protocol, an aqueous heparin solution (inner water phase) is first emulsified into a polymer/organic solution (oil phase), and this primary emulsion is then dispersed into a second aqueous phase containing a stabilizer such as poly(vinyl alcohol) (PVA). Upon solvent removal, solid microspheres are formed with heparin predominantly localized in internal water-rich domains. Indeed, heparin-loaded microspheres produced by double emulsion often exhibit a core–shell morphology, with the hydrophilic drug concentrated in internal reservoirs or pockets [109]. Several studies have successfully implemented this double emulsion method for heparin encapsulation. For example, Hoffart et al. formulated low-molecular-weight heparin (LMWH)-loaded PLGA/PCL microspheres via a W/O/W process, achieving encapsulation efficiencies of ~16–47% (enhanced by the inclusion of cationic polymers) and showing that released heparin retained its anticoagulant activity (anti-factor Xa). In another study, Chung et al. immobilized heparin on porous PLGA microspheres to bind basic fibroblast growth factor (bFGF); the heparin-functionalized microspheres markedly reduced the initial burst and provided sustained bFGF release, resulting in significantly enhanced angiogenesis in vivo compared to non-heparin controls [110]. One recent example of heparin-loaded microspheres produced by the emulsification–solvent evaporation method is reported by Rajabi et al. [95]. The authors demonstrated that this biocompatible microsphere system could sustain the anticoagulant’s release over an extended period, which is beneficial for tissue engineering applications requiring long-term heparin activity. In their study, poly(L-lactic acid) (PLLA) microspheres encapsulating heparin were fabricated via a W/O/W emulsification process, providing a reliable strategy for prolonged heparin release. These examples highlight that the emulsion method can preserve heparin–GF interactions and enable tunable release kinetics through formulation choices (e.g., incorporating charge complexes or targeting a core–shell structure). While the double emulsion technique generally yields high encapsulation efficiency and prolonged release for heparin, one challenge is the leakage of hydrophilic drug into the external aqueous phase during emulsification, which can reduce loading and must be mitigated (for instance, by optimizing phase composition or using osmotic agents) [107].

#### 3.1.2. Spray Drying

Spray drying is a rapid, scalable, and industrially favored method for fabricating dry microspheres by atomizing a polymer–drug solution into hot gas, which immediately evaporates the solvent and solidifies the droplets into particles [111]. In the context of heparin-loaded microspheres, spray drying offers a one-step, versatile route to encapsulate heparin either as a free molecule or as part of a complex (for example, heparin–protein or heparin–polycation complexes) [96]. Because heparin is hydrophilic and heat-sensitive, spray-drying parameters are often adjusted to mild conditions—for instance, using low inlet temperatures (around 90–120 °C)—and including protective excipients or hydrophilic carrier matrices (such as chitosan, gelatin, or trehalose) to preserve bioactivity [112]. The resulting spray-dried heparin microspheres typically have a smooth, spherical morphology and readily rehydrate in aqueous media, making them suitable for applications like pulmonary inhalation, topical delivery, or reconstitutable injectables [113]. Notably, spray drying also enables co-encapsulation of GFs or cytokines alongside heparin, capitalizing on heparin’s strong electrostatic binding to stabilize these biomolecules and modulate their release profiles [114]. For example, Yang et al. encapsulated heparin into PLGA microspheres via spray drying to create a prolonged-release system; the particles exhibited a triphasic release (with only an initial 24 h burst), extending heparin release for ~2 weeks, and the released heparin remained biologically active in suppressing vascular smooth muscle cell proliferation (comparable to native heparin) [96]. In a more recent example, Yildiz et al. developed inhalable heparin-loaded PLGA microspheres by spray drying and demonstrated that, when nebulized, these microparticles achieved sustained heparin release over 8–12 h in vitro and produced a more prolonged therapeutic effect on airway hyper-reactivity and inflammation in an asthma model compared to nebulized heparin solution. A state-of-the-art example of spray-dried heparin microspheres was given by Anaya et al. They utilized a 3D-printed micromixer combined with spray drying to prepare inhalable microparticles co-loaded with heparin and azithromycin [97]. The resulting dry powder microspheres exhibited potent anti-inflammatory effects and were capable of inhibiting both SARS-CoV-2 and bacterial pathogens in the lung. These studies illustrate the promise of spray-dried heparin microspheres in preserving heparin’s function and enhancing its duration of action. Nevertheless, despite its advantages, spray drying can subject encapsulated biomolecules to thermal and shear stresses during atomization and drying, which may risk denaturation or loss of activity, especially for proteins [115].

#### 3.1.3. Microfluidics

Microfluidics offers a highly precise and controllable platform for fabricating monodisperse microspheres through the manipulation of fluids in microscale channels [116]. In contrast to conventional bulk emulsification methods, microfluidic droplet synthesis provides exceptional control over droplet size, uniformity, shape, and composition, which is especially attractive for heparin-loaded microspheres where reproducible structure and consistent release behavior are critical. In a typical droplet-based microfluidic system, immiscible fluid streams (for example, an aqueous phase and an oil phase) are introduced into a microchannel device such that shear forces or flow-focusing at junctions break the flow into discrete, monodisperse droplets. These droplets can subsequently be solidified or cross-linked (ionically, chemically, or via photo-polymerization) on-chip or after collection, yielding stable microspheres. For instance, water-in-oil (W/O) or water-in-oil-in-water (W/O/W) double emulsion droplets can be generated in microfluidic devices for encapsulating hydrophilic agents like heparin in a single step. Heparin-containing microspheres produced via microfluidics often integrate advanced functionalities, such as core–shell architectures for sequential or staged release, co-encapsulation of GFs, or surface-presented bioactive cues to facilitate cell interactions. Ma et al. demonstrated a hierarchical core–shell design using microfluidic double emulsions: heparin-conjugated gelatin nanospheres loaded with BMP-2 were encapsulated within larger PLGA microspheres, resulting in a multi-phase delivery system where the GF release was regulated both by heparin binding and by the microsphere’s structure [117]. This heparin-based system provided sustained release of BMP-2 and significantly enhanced in vivo bone regeneration compared to conventional microspheres. In another example, Young et al. employed a capillary microfluidic device with UV photopolymerization to produce uniformly sized hydrogel microspheres (200–800 μm) from PVA and heparin [116]. The resulting biosynthetic microspheres had mechanical properties comparable to bulk hydrogels and supported over 90% cell viability for 4 weeks in culture, underscoring their potential for cell encapsulation and tissue engineering applications. Additionally, microfluidic techniques have been used to fabricate microspheres from mixtures of photoreactive precursors—for example, combining heparin methacrylate (HepMA) with poly(ethylene glycol) diacrylate (PEGDA) in a microchannel, followed by UV crosslinking of the droplets into heparin-functionalized microgels with tunable degradation and release properties [100]. Emerging microfluidic techniques have also been applied to create heparin-functionalized microspheres with high uniformity. For instance, Lei et al. developed chitosan-based hydrogel microspheres using a droplet microfluidic approach, aiming to improve wound healing [118]. These microspheres were designed as multifunctional carriers that could capture and release GFs in one step, leveraging heparin-mimicking interactions to bind therapeutic proteins and enhance tissue repair. While the throughput of traditional microfluidic single-channel droplet generators is relatively low, recent advances are addressing this limitation. Parallelized droplet microfluidic arrays can massively increase production rates—for instance, using multiple synchronized nozzles has been shown to boost microsphere output by ~600% without sacrificing uniformity [99]. Moreover, emerging fabrication approaches such as 3D-printed droplet microfluidic devices are improving scalability and robustness, bringing microfluidic production of specialized microspheres closer to clinical and industrial feasibility.

#### 3.1.4. Coacervation

Coacervation (phase separation) is a microencapsulation technique in which a polymer-rich phase separates out of solution and wraps around a dispersed drug to form microspheres [119]. Coacervation is a phase-separation technique that has long been employed to create polymer-rich microdroplets for encapsulation [89]. In simple coacervation, a single polymer is induced to separate (e.g., by adding a non-solvent or changing pH), whereas complex coacervation involves two oppositely charged polymers forming a dense colloidal phase via electrostatic attraction. This method offers high loading efficiency under mild conditions, which helps preserve the bioactivity of sensitive therapeutics [101]. For hydrophilic drugs like heparin, complex coacervation is especially relevant: the polyanionic heparin can be paired with cationic polymers (gelatin, chitosan, polyamines, etc.) to coacervate into microspheres that entrap heparin in a polymer-rich matrix. For example, Mouftah et al. prepared cationic polymethacrylate nanoparticles via coacervation to deliver low-molecular-weight heparin through the buccal route, achieving effective mucoadhesion and sustained release [120]. For example, Gao et al. used a heparin/PEAD complex coacervate as a sustained-release depot for bone morphogenetic proteins (BMP-2, -4, -6, -7, -9) in a calvarial bone defect model [102]. In that system, up to 2 μg of BMP could be loaded into the heparin-based coacervate, and BMP-2 emerged as the most potent, regenerating ~10× more new bone than other BMPs when delivered via coacervate. However, because coacervate droplets are liquid and membrane-free, additional steps like chemical crosslinking or coating are often needed to stabilize the microspheres and prevent them from coalescing or leaking payload. Recent advances (e.g., integrating bio-inspired tannic acid coatings) have shown that it is possible to stabilize coacervate microdroplets without compromising their permeability, addressing a key challenge for translating this technique [75]. Together, these examples demonstrate that coacervation can effectively encapsulate heparin (often alongside GFs or antioxidants), preserve critical heparin–protein interactions, and enable tunable, sustained release profiles. While coacervation generally achieves high heparin loading and prolonged activity, careful formulation (e.g., optimizing phase ratios or adding stabilizers) is required to minimize initial burst release and ensure uniform, stable microspheres [119].

#### 3.1.5. Electrospraying

Electrospraying (electrohydrodynamic atomization) is a technique that generates fine droplets by applying a high voltage to a polymer solution as it passes through a needle, forming a charged jet that breaks into micro- or nanoscale droplets [121]. Unlike conventional pneumatic spraying or heat-based methods, electrospraying can produce highly uniform particles at ambient temperature, making it suitable for encapsulating delicate biomolecules like heparin without thermal stress. The droplet size and morphology can be tuned by adjusting parameters such as solution conductivity, flow rate, and voltage. Notably, electrospraying can be configured in a coaxial setup to create core–shell microspheres in one step. For instance, Li et al. fabricated core–shell PCL/PEG microspheres with a heparin-rich core via coaxial electrospraying and deposited these onto vascular graft surfaces to create an anti-thrombogenic coating [103]. Similarly, Mao et al. optimized coaxial electrospraying conditions (polymer concentration, voltage, flow rate, etc.) for heparin/PEG-PCL core–shell carriers; they confirmed the core–shell structure by TEM and showed that varying the shell polymer content controlled the heparin encapsulation efficiency and release rate [104]. Electrospraying has also been applied to formulate inhalable heparin-containing particles. Anaya et al. reported heparin–azithromycin composite microparticles (for pulmonary delivery) produced by electrospraying, which exhibited potent anti-inflammatory effects and were capable of inhibiting SARS-CoV-2 as well as bacterial pathogens in the lungs [97]. Moreover, multi-drug encapsulation is feasible: coaxial electrospraying has been used to create PLGA core–shell microspheres for sequential release of two drugs (e.g., paclitaxel and imatinib) in combination therapy, underscoring the method’s flexibility in designing combination delivery systems (though this particular case did not involve heparin, it highlights the technique’s capability) [105]. Electrospraying can also produce hydrogel microspheres suitable for cell encapsulation. Because the process is gentle and rapid, cells or proteins can be encapsulated with minimal loss of viability/activity. For example, electrosprayed PVA–heparin hydrogel microspheres have shown over 90% cell viability 24 h post-fabrication, indicating the technique’s cell-friendly nature [116]. In another demonstration, Wu et al. used a microfluidic electrospray approach to generate porous alginate microspheres encapsulating living NK cells for tumor treatment—the monodisperse microspheres maintained cell functionality and provided a protective carrier for systemic delivery [98]. In one 2025 report, core–shell microspheres loaded with an antioxidant peptide and fibronectin (FN) were generated via coaxial electrospraying; these “RPG@FN” particles were shown to synergistically reduce inflammatory responses by scavenging reactive oxygen species (ROS) and polarizing macrophages toward a reparative M2 phenotype. Such immunomodulatory microspheres, made possible by the co-loading capabilities of electrospraying, can create a pro-healing microenvironment and have been proposed for applications like chronic wound or disc degeneration therapy [122]. Overall, electrospraying offers excellent control over particle size and architecture and has proven effective for encapsulating heparin in forms ranging from anticoagulant microspheres to multi-functional drug delivery particles.

#### 3.1.6. Summary

Heparin-based microspheres are injectable, possess high surface-area-to-volume ratios for efficient GF loading and release, and support cell adhesion—making them ideal for minimally invasive delivery and use as micro-scaffolds in tissue engineering [123,124]. Despite these advantages, conventional microspheres have some limitations. A major challenge is controlling their degradation and release rates—for example, in vivo degradation of PLGA microspheres can be difficult to tune precisely, potentially leading to an initial burst release of GF cargo [123]. To address these issues, researchers have developed advanced microsphere designs—including macroporous, core–shell, and nanofibrous microspheres—which improve control over release kinetics and enhance cell interactions [121]. Moreover, since injected microspheres are discrete particles lacking a continuous structure, the mechanical integrity at the implantation site may be weak. Embedding microspheres into a hydrogel matrix or crosslinking them in situ can form a composite scaffold that prevents particle migration and enhances mechanical cohesion. Notably, incorporating biodegradable microspheres into hydrogels has been shown to significantly enhance the construct’s mechanical strength (in one study, >100-fold increase in compressive modulus) [125], while also improving cell adhesion and osteogenic differentiation compared to hydrogel alone. Such composite systems leverage the injectability and high loading capacity of microspheres together with the structural support of hydrogels, thereby overcoming the stand-alone drawbacks of microsphere carriers.

### 3.2. Heparin-Loaded Nanofibers and Fabrication Strategies

Nanoscale fibrous scaffolds have gained increasing attention in regenerative medicine due to their ability to mimic the native ECM in both architecture and function. Their high surface-to-volume ratio, tunable porosity, and flexibility in material selection make them ideal platforms for incorporating bioactive molecules such as GFs [126,127]. Among these, heparin-functionalized nanofibers represent a particularly promising class of delivery systems. Heparin, a highly sulfated glycosaminoglycan, can non-covalently bind a wide range of GFs via electrostatic interactions, stabilizing their structure and prolonging their biological activity in vivo [128,129]. By integrating heparin into nanofiber matrices, researchers have developed systems capable of sustained and localized GF delivery. One effective approach involves embedding heparin directly within the fiber bulk or immobilizing it on the surface, enabling controlled interactions with target proteins. For example, Ikegami et al. fabricated aligned poly(ε-caprolactone)/gelatin nanofibers with covalently immobilized heparin to achieve long-term retention of bFGF and NGF, promoting peripheral nerve regeneration in vivo [128]. Similarly, Qiu et al. incorporated heparin into an elastic PU matrix using emulsion blending, resulting in a tubular vascular scaffold with improved hemocompatibility and enhanced endothelialization following implantation [129]. These findings demonstrate that even simple uniaxial fiber configurations, when combined with heparin’s biomolecular affinity, can yield delivery systems with prolonged GF retention, reduced burst release, and improved therapeutic efficacy across various tissue engineering applications. A variety of electrospinning strategies have since been developed to further optimize heparin integration, drug release profiles, and structural complexity. Table 4 provides a comparative overview of key electrospinning approaches—including coaxial, triaxial, emulsion, Janus, and multilayer concentric configurations—highlighting their respective advantages, limitations, and potential for heparin-based fibrous drug delivery systems.

#### 3.2.1. Coaxial Electrospun Core–Shell Nanofibers

Core–shell nanofibers fabricated via coaxial electrospinning offer a versatile platform for spatially controlled encapsulation and sequential release of bioactive molecules [141]. This technique employs a dual-capillary spinneret to simultaneously extrude distinct core and sheath polymer solutions, resulting in nanofibers with well-defined compartmentalization. Both heparin and GFs can be selectively localized in either the core or shell, enabling the independent tuning of release kinetics, degradation behavior, and biological interactions [126,130]. One representative study by Fahad et al. reported the fabrication of a small-diameter vascular graft using coaxial electrospinning, in which a PCL core was encapsulated by a gelatin sheath containing a heparin–VEGF complex. The gelatin layer provided rapid degradation-mediated release of approximately 80–86% of both heparin and VEGF over 25 days, while the PCL core maintained structural integrity. In vivo evaluation in a rat abdominal aorta model demonstrated complete endothelialization and robust smooth muscle cell regeneration, achieving 100% patency after implantation [130]. In another example, Joshi et al. designed coaxial nanofibers with a heparin-loaded PCL core and gelatin shell. The inner core acted as a reservoir for sustained heparin release, which not only prolonged the presence of exogenous factors but also enhanced the adsorption of endogenous angiogenic proteins. This dual-action mechanism resulted in significantly improved wound healing outcomes compared to control fibers lacking heparin [131]. The key advantage of coaxial electrospinning lies in its ability to decouple mechanical and biochemical functions. The shell—often composed of hydrophobic polymers such as PCL—provides tensile strength and load-bearing capacity, while the core—made from hydrophilic or biodegradable polymers like gelatin—facilitates drug diffusion, factor loading, and biointerfacing [132]. Moreover, this approach significantly reduces burst release, as the outer shell acts as a diffusion barrier that slows the outward migration of encapsulated agents. Heparin, in particular, benefits from this architecture, as it can be localized in the shell for immediate interaction with host tissues or in the core for long-term release. Additionally, coaxial systems allow the co-encapsulation of multiple drugs in discrete compartments, enabling synergistic or sequential therapy. Despite these advantages, coaxial electrospinning presents several technical challenges [142]. The process requires precise control of flow rates, solvent compatibility, and solution viscosities to maintain a stable core–shell jet. Improper parameter tuning can lead to jet instability, nozzle clogging, core discontinuity, or delamination at the core–shell interface. Moreover, the specialized spinneret configuration and low throughput of the process limit its scalability for industrial production. Incompatibility between the encapsulated drug and its carrier matrix may still result in premature leakage, especially if the sheath material dissolves rapidly under physiological conditions. Compared to conventional uniaxial or blend electrospinning, coaxial fibers consistently demonstrate superior encapsulation efficiency and more sustained, tunable release profiles. For example, in a comparative study, a dual-agent core–shell fiber system achieved more complete and independent release of two active compounds compared to a single-nozzle blended fiber, highlighting the importance of spatial compartmentalization.

#### 3.2.2. Microfluidic and Emulsion Electrospinning

Pseudo-coaxial strategies, including microfluidic-assisted electrospinning and emulsion electrospinning, offer alternative routes to fabricate core–shell-like nanostructures without the need for specialized coaxial nozzles. These approaches are particularly useful for integrating labile biomolecules such as heparin and GFs, as they often provide a gentler fabrication environment and simplified operational setup.

Microfluidic-Assisted Electrospinning

Microfluidic spinning systems utilize microchannels or parallel capillaries to guide multiple polymer flows under laminar conditions before jetting. This enables the formation of structured fibers with tunable core–shell or multilayered geometries. For example, Zhang et al. developed a microfluidic setup that continuously extruded alginate as a core and a PCL-based drug-loaded solution as a sheath, forming aligned core–shell fibers embedded with nanoparticles [143]. In a study by Vasconcelos et al., PCL/PVA meshes fabricated via microfluidic-assisted electrospinning were used to co-deliver methotrexate and an anti-TNF-α antibody. These dual-loaded fibers exhibited higher local drug retention and enhanced anti-inflammatory activity compared to those produced using traditional coaxial electrospinning [133]. In a particularly relevant example, Cheng et al. employed microfluidic-assisted electrospinning to deposit heparin and VEGF directly onto the luminal surface of vascular stents, while an outer layer contained rapamycin [133]. This hierarchical fiber arrangement allowed simultaneous anticoagulation, endothelialization, and inhibition of smooth muscle proliferation, highlighting the precision and modularity of microfluidic systems in therapeutic fiber design [134]. Advantages of microfluidic-assisted electrospinning include precise control over polymer flow rates, the ability to construct highly uniform fiber structures, and compatibility with in-line crosslinking methods such as UV curing. The modular nature of microfluidic systems also facilitates continuous production and integration into more scalable platforms. However, limitations include technical complexity in device fabrication, susceptibility to clogging, and the need for finely tuned pressure control systems to maintain stable flows.

Emulsion Electrospinning

Emulsion electrospinning provides a simpler alternative for producing pseudo-core–shell fibers using only a single nozzle. In this technique, an aqueous solution containing heparin or GFs is emulsified into an organic polymer phase, typically forming water-in-oil (W/O) droplets stabilized by surfactants. Upon electrospinning, the continuous polymer matrix solidifies while the aqueous droplets become discrete, encapsulated “microcores” within the fibers [144]. This technique offers indirect compartmentalization and protects sensitive agents from organic solvents. Liu et al. successfully applied emulsion electrospinning to fabricate PLGA-based nanofibers loaded with glial cell line-derived neurotrophic factor (GDNF) in the dispersed phase. The resulting fibers demonstrated extended GDNF release with preserved biological activity, significantly promoting nerve regeneration in preclinical models [135]. In the vascular field, Zhang et al. incorporated heparin into a polyurethane-based emulsion system to create heparin–polyurethane hybrid nanofibers (H-PHNF). These fibers exhibited sustained heparin release, low thrombogenicity, and favorable endothelial cell compatibility, making them highly relevant for vascular graft applications [136]. The advantages of emulsion electrospinning include operational simplicity (single-needle setup), suitability for sensitive biomolecules (due to limited direct solvent exposure), and the ability to achieve biphasic or multi-phase release by adjusting droplet size and emulsion stability. On the other hand, limitations include potential polydispersity in internal droplet sizes, irregular microstructure, and reduced mechanical strength of the final fiber mats due to embedded pores. Additionally, precise control over the spatial distribution of encapsulated agents is more difficult compared to coaxial or microfluidic methods.

#### 3.2.3. Triaxial, Janus, and Multilayer Concentric Nanofibers

To achieve multi-component delivery with fine-tuned spatial and temporal control, researchers have developed advanced nanofiber designs—triaxial, Janus (side-by-side), and multilayer concentric fibers. These configurations build on the core–shell concept, enabling sophisticated release profiles and co-delivery of heparin and GFs in a single fiber.

Triaxial Electrospinning

Triaxial electrospinning extends the principle of coaxial electrospinning by employing three concentric fluid streams—typically a central core, an intermediate layer, and an outer shell [145,146]. Each layer may carry a different polymer or bioactive compound. For example, the innermost core might contain heparin and a GF (e.g., VEGF), the intermediate layer a secondary factor or diffusion barrier, and the outer shell a mechanically supportive or protective material [137,138]. Such compartmentalization offers enhanced control over both mechanical properties and release kinetics. A gradient release strategy can be achieved by tuning the degradation rates of each layer, enabling early release of one factor and delayed release of another. Triaxial constructs have been shown to improve encapsulation efficiency, reduce initial burst release, and enhance biological outcomes such as angiogenesis and neurogenesis. Moreover, the additional layer contributes to better structural stability, which is advantageous for implantable scaffolds. However, the fabrication process is technically demanding, requiring precise flow alignment and viscosity matching, and suffers from relatively low throughput.

Janus (Side-by-Side) Electrospinning

Janus fibers are fabricated using parallel spinning nozzles that extrude two separate solutions side by side. The resulting fiber has two distinct faces, each composed of a different polymer or bioactive formulation [147,148]. This architecture is especially useful for combining incompatible materials or delivering agents with distinct release profiles. For example, one side of a Janus fiber can consist of a hydrophilic polymer containing heparin or a GF, while the other comprises a hydrophobic polymer loaded with a small-molecule drug. Sun et al. demonstrated a tri-layer Janus fiber with polyvinylpyrrolidone (PVP), PVP–cellulose acetate, and cellulose acetate segments, designed for asymmetric multi-drug release [139]. Such constructs enable biphasic release profiles, with one side rapidly dissolving to release an initial dose, while the other degrades slowly to provide prolonged exposure. Janus fibers also allow for spatial presentation of functional groups—for instance, heparin or adhesion peptides can be confined to one face for targeted biointeraction. However, maintaining jet stability during side-by-side spinning can be challenging, and fiber uniformity may suffer if flow rates or viscosities are not perfectly matched [142].

Multilayer Concentric Electrospinning

The term “multilayer concentric fibers” encompasses coaxial, triaxial, and more complex architectures with multiple concentric polymer shells. The primary advantage of such designs is their potential for highly tailored spatiotemporal control of drug release. By carefully selecting polymers for each layer—varying in hydrophilicity, degradation rate, and diffusion barrier properties—engineers can fine-tune when and how each agent is released. For instance, a multilayer fiber might include an innermost reservoir of heparin-bound GFs, an intermediate polymer layer to delay diffusion, and an outer shell providing structural and mechanical support [140]. These systems mitigate burst release and better protect fragile proteins such as VEGF and bFGF during fabrication and deployment. While effective, their development is limited by fabrication complexity, potential interfacial instability, and low production scalability.

#### 3.2.4. Another Nanofibers

To further expand the applicability of heparin-functionalized nanofibers in clinical translation, melt electrospinning has garnered particular attention for its potential in producing high-throughput or solvent-free scaffolds while retaining functional bioactivity. Melt electrospinning, which eliminates solvents and enables precise fiber deposition, is advantageous for 3D scaffold design. Yet, its high temperatures are incompatible with thermolabile agents like VEGF. Zhang et al. addressed this by post-loading VEGF onto melt-electrowritten PCL scaffolds via plasma-assisted conjugation, preserving activity while avoiding thermal damage [149]. Still, the added complexity may limit scalability. These advanced methods broaden the design space for heparin-based delivery, though practical barriers to protein stability and process control remain.

While electrospinning remains the dominant method for fabricating heparin–GF-loaded nanofibers [150,151], alternative fiber-forming strategies such as wet spinning, freeze-drawing, and solution blow spinning have shown promise in other biomedical contexts but have not yet been reported for this specific combination. This gap likely stems from the challenge of simultaneously encapsulating fragile, low-dose biologics in scalable or thermal-based fiber processes [152,153]. Nevertheless, cold-processing methods such as cryo-spinning or directional freeze-drying could offer solvent-free, mild alternatives to preserve GF bioactivity while enabling fiber formation. Developing such approaches represents an underexplored opportunity to expand heparin-based delivery platforms beyond conventional electrospinning.

#### 3.2.5. Summary

Electrospun nanofibers are a widely used heparin-functionalized platform due to their biomimetic architecture and ease of functionalization with GF-binding molecules [29]. Their material versatility and alignment potential make them suitable for directing cell behavior and tissue organization. Despite their advantages, electrospun nanofiber scaffolds face several limitations. First, traditional densely packed mats with micron-scale pores hinder cell infiltration and nutrient diffusion; to address this, researchers have introduced microfibers or sacrificial porogens during fabrication to generate multi-scale fiber networks with larger interconnected pores, thereby enhancing oxygen transport and cellular ingress [154]. Second, nanofiber sheets in their solid form are not injectable, limiting their applicability in minimally invasive procedures; this challenge has been mitigated by fragmenting nanofibers into short fibrils and incorporating them into shear-thinning hydrogels, which allows for injection and subsequent gelation at the target site [155,156]. Lastly, conventional nanofiber scaffolds often support only a single cell type due to their uniform architecture; this issue is being tackled by constructing composite scaffolds or co-culture systems that better recapitulate the complexity of native tissues [157].

### 3.3. Heparin-Based Hydrogel Platforms

Given the breadth of hydrogel research and the frequent use of crosslinking strategies as a foundational element in their design, a vast array of review articles has already addressed various aspects of hydrogel systems. Rather than repeating these established summaries, this section begins by recommending a selection of recent and comprehensive reviews that focus on different themes within the field of heparin-based hydrogels.

For those seeking to understand the design and application of heparin-functionalized hydrogels, the review by Gupta et al. [30]. provides a comprehensive overview of chemical strategies for heparin immobilization, highlighting its biomedical relevance in drug delivery, anticoagulation, and GF retention.

To explore the use of heparin as a mediator for GF delivery, the foundational work by Abune and Wang [158] is recommended. This article reviews affinity-based hydrogel platforms that incorporate heparin or heparin-mimicking ligands for protein sequestration and controlled release.

Readers interested in hydrogel synthesis and crosslinking strategies—including physical, ionic, covalent, and dynamic chemistries—should consult the authoritative review by Hu et al. [159], which categorizes crosslinking mechanisms and matches them to application contexts such as tissue scaffolding and drug delivery.

For insights into injectable and in situ-forming heparin-based hydrogels, the work of Bertsch et al. [160] offers a detailed analysis of self-healing, shear-thinning materials that allow minimally invasive deployment in regenerative medicine.

For an overview of multifunctional composite hydrogels, see Yuan et al. [161], which discusses the integration of drugs, nanomaterials, and cells into hydrogels to enhance antibacterial, hemostatic, and regenerative properties.

Lastly, injectable hydrogels in cartilage and bone repair, Ghandforoushan et al. [162] reviews recent advances in in situ-forming systems designed for minimally invasive musculoskeletal tissue regeneration.

## 4. Stimuli-Responsive Designs and Therapeutic Applications of Heparin-Based GF Delivery Platforms

### 4.1. Functional Designs: Stimuli-Responsive Delivery Systems

Heparin-based delivery platforms have evolved beyond passive drug carriers to become dynamic systems capable of responding to specific physiological or pathological stimuli. This functional adaptability is particularly crucial in the context of GF delivery, where spatiotemporal precision can significantly influence therapeutic outcomes. Stimuli-responsive systems are engineered to undergo structural or chemical changes in response to environmental cues, such as pH shifts, temperature variations, enzymatic activity, redox imbalances, or externally applied forces like ultrasound. These triggers can be harnessed to control the release kinetics, enhance targeting, and minimize off-target effects of heparin-bound therapeutics. By integrating stimulus-sensitive components with heparin’s natural affinity for GFs, such platforms enable “smart” release behavior, improving bioavailability and therapeutic efficacy in complex biological environments. The following subsections explore major categories of stimuli-responsive heparin-based delivery systems and provide recent examples demonstrating their mechanism, performance, and biomedical relevance.

#### 4.1.1. pH-Responsive Heparin-Based Systems

Heparin-based nanocarriers can be rationally designed to respond to the acidic microenvironments characteristic of pathological sites such as tumors, inflamed tissues, or endosomes. Such pH-responsive systems often leverage acid-labile linkers or the ionizable functional groups inherent to heparin, enabling selective drug release under low pH conditions. For instance, Roberts et al. demonstrated that PVA–heparin hydrogels remained stable at physiological and acidic pH but degraded rapidly under basic conditions, enabling controlled release behavior based on environmental pH fluctuations [163]. In another study, She et al. engineered dendronized heparin–doxorubicin nanoparticles that exhibited significantly enhanced drug release at pH 5.0 compared to pH 7.4, achieving potent antitumor effects while minimizing toxicity to healthy tissues [164]. Heparin has also been employed to enhance the pH sensitivity of composite systems: for example, heparin-coated mesoporous silica nanoparticles loaded with platinum-based drugs released their cargo more rapidly at pH 5.8 than at neutral pH, attributed to the protonation of heparin carboxyl groups facilitating matrix destabilization [165]. However, it is important to note that while pH-triggered systems can be effective for small-molecule drugs, they may not be universally suitable for sensitive biologics such as GFs. Due to their delicate tertiary structures, many GFs are prone to denaturation or loss of bioactivity under acidic conditions, limiting the applicability of low pH-triggered release mechanisms for such proteins. Collectively, these examples highlight how the chemical versatility of heparin can be leveraged to construct pH-sensitive carriers for site-specific and environmentally triggered delivery.

#### 4.1.2. Thermo-Responsive Systems

Heparin-functionalized systems can be integrated into thermosensitive polymers to yield injectable, in situ gelling platforms that respond to physiological temperature changes. One widely used approach involves Pluronic (poloxamer) or poly(N-isopropylacrylamide) (PNIPAM) polymers, which undergo sol–gel transitions near body temperature. For example, a heparin–poloxamer 407 hydrogel was developed to remain liquid at room temperature but form a stable gel at 37 °C, enabling the co-delivery of bFGF and NGF for peripheral nerve regeneration [166]. Similarly, Roberts et al. demonstrated that grafting heparin into a PVA hydrogel framework significantly reduced its thermal stability; while intact for months at 25 °C, the material degraded within 4 days at 50 °C, underscoring the tunable thermal responsiveness imparted by heparin [163]. In another study, thermo-responsive polyelectrolyte multilayers composed of PNIPAM and heparin were shown to undergo reversible swelling and collapse at temperatures around 31–33 °C, facilitating controlled loading and release of proteins such as VEGF and bFGF during temperature cycling [167]. Heparin also modulates the thermal transition behavior of composite systems: Chung et al. observed that incorporation of heparin into Pluronic F127 hydrogels decreased the gelation temperature, attributed to electrostatic and hydrophobic interactions between the polymer and heparin chains [168]. Overall, thermo-responsive heparin-based systems offer precise temporal control over GF delivery while maintaining the advantages of injectability, biocompatibility, and localized gelation at physiological temperature.

#### 4.1.3. Enzyme-Responsive Systems

Heparin-based delivery platforms can be engineered to respond to enzymatic activity, particularly by integrating protease-cleavable linkages that enable site-specific payload release in response to biological remodeling. A prominent approach involves incorporating matrix metalloproteinase (MMP)–sensitive peptides into the carrier architecture. For instance, sensitive peptides yielded injectable starPEG–heparin hydrogels that disassembled in response to protease activity, making them suitable for dynamic tissue environments [169]. Yang et al. reported a VEGF-loaded hydrogel composed of heparin-mimicking polymers crosslinked by MMP-cleavable sequences; in ischemic limbs, MMP-mediated hydrogel degradation enabled controlled VEGF release, promoting vascularization and anticoagulation [170]. Enzymatic responsiveness also applies to intracellular settings. Dogru et al. demonstrated that in chondrocyte cultures, TGF-β bound to a heparin-conjugated scaffold was released 160–360× faster than in acellular systems due to cell-mediated uptake and proteolysis, confirming the role of endogenous proteases in GF liberation [171]. Together, these studies reveal that heparin-functionalized hydrogels can be precisely tailored to degrade in the presence of extracellular or intracellular enzymes, offering on-demand release of GFs in wound healing, tumor therapy, and regenerative medicine.

#### 4.1.4. ROS/Redox-Responsive Systems

Heparin-based delivery platforms can be sensitively engineered to respond to redox environments, such as elevated intracellular glutathione (GSH) levels or pathological concentrations of ROS. A common redox-responsive mechanism involves the incorporation of disulfide bonds, which are stable extracellularly but cleaved in reductive intracellular conditions. For example, a heparin-mimicking hydrogel crosslinked with ROS-sensitive boronic esters was shown to release both an antioxidant and VEGF only under inflammatory ROS conditions, achieving inflammation-targeted delivery [170]. In another case, a cisplatin-loaded Hep-P407 nanogel demonstrated selective drug release under reducing conditions, which improved tumor cell cytotoxicity while minimizing off-target effects [172]. Furthermore, a redox-degradable heparin hydrogel system was designed for GF delivery in oxidative microenvironments, enhancing angiogenesis only where ROS levels were pathologically elevated [173]. Collectively, these findings demonstrate that heparin-based carriers incorporating redox- or ROS-sensitive moieties provide controlled, stimulus-triggered release of bioactive agents, particularly useful for targeting tumor, ischemic, or inflamed tissues.

#### 4.1.5. Ultrasound-Responsive Systems

Heparin-based platforms have been effectively integrated with ultrasound (US)-responsive delivery strategies, particularly through the development of heparin-loaded microbubbles (HPMBs). These HPMBs—typically gas-filled vesicles coated or embedded with heparin—circulate stably in the bloodstream but rupture upon exposure to focused ultrasound, thereby releasing their payload at the targeted site. In a rat model of pancreatitis, HPMBs released their heparin content upon ultrasound insonation, leading to a significant reduction in oxidative stress and inflammatory responses in the diseased tissue [174].

### 4.2. Applications of Heparin-Based GF Delivery Platforms

Heparin-based delivery platforms not only offer versatile structural designs but also demonstrate significant therapeutic potential across a broad spectrum of regenerative and pathological conditions. Building upon their capacity to stabilize GFs, prolong bioactivity, and provide localized release, these platforms have been successfully employed in diverse biomedical applications. This section outlines the key therapeutic domains where heparin-functionalized carriers have shown promise, including soft tissue and musculoskeletal regeneration, neural repair, cardiovascular healing, chronic wound management, anti-fibrotic therapies, and tumor microenvironment (TME) modulation, as shown in Figure 3. Through these case-specific implementations, the clinical relevance and translational potential of heparin-based delivery systems are further underscored.

#### 4.2.1. Soft Tissue Regeneration

Heparin-based nanocarriers have been widely explored for promoting soft tissue regeneration, particularly by modulating the localized delivery of pro-regenerative GFs such as EGF, FGF, VEGF, PDGF, and BMP. In cutaneous wound healing, heparin–coacervate systems have shown outstanding regenerative potential. Johnson and Wang reported that coacervates formed from heparin and polycation, loaded with heparin-binding EGF (HB-EGF), significantly enhanced keratinocyte migration and angiogenesis, accelerating re-epithelialization far more efficiently than free HB-EGF alone [178]. In the context of skeletal muscle repair, He et al. developed a heparinized decellularized ECM scaffold enriched with autologous platelet-rich fibrin (containing PDGF, FGF, and VEGF). The heparin component enabled sustained release of these GFs, effectively enhancing endothelialization and myogenesis in a rat muscle defect model [175]. For tendon–bone interface repair, silk scaffolds functionalized with heparin and loaded with TGF-β2 and GDF5 were shown to maintain GF bioactivity even at low doses. These systems promoted site-specific differentiation of mesenchymal stem cells into enthesis-like tissue, improving the mechanical performance of the healing site [179]. Heparin microparticles serve as another versatile delivery platform. Bromberg et al. demonstrated that BMP-2-loaded heparin microparticles not only stabilized large quantities of the GF but also matched the osteogenic effects of high-dose soluble BMP-2, thereby illustrating their value for localized tissue regeneration [93]. Finally, composite scaffolds combining heparin with ECM materials, such as collagen, have shown synergistic pro-angiogenic effects. For instance, a collagen/heparin scaffold incorporating FGF2 and VEGF successfully induced neovascularization and improved skin regeneration in a fetal wound model [180]. Together, these findings confirm that heparin’s GF affinity and release-controlling capabilities can be effectively used to restore soft tissue structure and function across a wide range of clinical applications.

#### 4.2.2. Bone and Cartilage Repair

Heparin-based delivery systems have shown great promise in enhancing bone and cartilage regeneration by localizing and sustaining osteogenic GFs such as BMP-2, TGF-β, and FGF. One classical strategy involves the modification of demineralized bone matrix (DBM) with heparin. Lin et al. developed heparin-conjugated DBM (HC-DBM), which exhibited significantly enhanced BMP-2 binding capacity and mechanical strength. When loaded with BMP-2, HC-DBM stimulated greater osteogenic activity—evident from higher alkaline phosphatase (ALP) activity and calcium deposition—compared to unmodified DBM in vivo [181]. In parallel, heparin microparticles have been used to localize BMP-2 within bone defects. These microparticles enable sustained retention and controlled release of osteogenic signals, inducing bone formation comparable to that achieved with high doses of soluble BMP-2 [93]. Heparin also supports endogenous GF therapy by enhancing the retention and bioavailability of bone-related GFs, such as BMPs, TGF-βs, and PDGF, within grafts or scaffolds. As Wang et al. have reviewed, this leads to improved angiogenesis and osteogenesis in large bone defects [182]. In tendon–bone interface tissue engineering, Spang et al. employed heparin-functionalized silk scaffolds to retain TGF-β2 and GDF5. These scaffolds enabled mesenchymal stem cells to differentiate into enthesis-like tissue structures, closely mimicking the native tendon-to-bone transition zone [179]. Despite these advances in bone regeneration, cartilage repair still presents challenges. Dogru et al. found that TGF-β, when delivered via heparin-conjugated scaffolds, was rapidly depleted due to cell-mediated uptake, reducing its therapeutic range in engineered cartilage constructs [171]. In summary, heparin-based systems substantially enhance bone repair by concentrating osteogenic GFs at defect sites and supporting their sustained bioactivity. While similar principles can be adapted for cartilage repair, design modifications are required to overcome challenges such as rapid cellular internalization of GFs.

#### 4.2.3. Neural Tissue Engineering

Heparin-based delivery systems have demonstrated significant potential in promoting peripheral and central nervous system (CNS) regeneration through the controlled release of neurotrophic factors such as nerve NGF, brain-derived neurotrophic factor (BDNF), and GDNF. In a foundational study, Wood et al. incorporated heparin-binding peptides into nerve guidance conduits to immobilize NGF. In vivo experiments using a rat sciatic nerve injury model showed that NGF delivered via this heparin-affinity platform significantly improved axonal density and fiber maturity compared to untreated controls [183]. Building upon this concept, Li et al. engineered a polycation–heparin coacervate that achieved nearly 100% NGF encapsulation efficiency. This coacervate not only protected NGF from degradation but also enabled a single injection post-injury to markedly enhance motor recovery and upregulate axonal and Schwann cell markers in the injured sciatic nerve [176]. Heparin also assists neurotrophic factor distribution within the brain. For example, during convection-enhanced delivery (CED) of GDNF-family ligands in rats, co-infusion of free heparin significantly improved tissue-wide dispersion of the therapeutic proteins. This was attributed to heparin’s ability to competitively block extracellular heparin-binding sites that would otherwise trap the GFs near the injection site [184]. Collectively, these studies show that heparin-affinity carriers and injectable coacervates can greatly enhance the stability, bioavailability, and spatial distribution of neurotrophic factors. As a result, they promote robust axonal growth and functional recovery in nerve injury models, offering a promising platform for neural tissue engineering.

#### 4.2.4. Cardiovascular Regeneration

Heparin-functionalized biomaterials have gained prominence in cardiovascular tissue engineering due to their dual ability to prevent thrombosis and sequester angiogenic GFs such as VEGF, FGF, and PDGF. A prime example involves the development of a GelMA/heparin-based 3D-printed bioink loaded with VEGF. This system enabled sustained VEGF release for over 21 days and facilitated endothelial network formation and capillary-like structures in vitro, demonstrating strong potential for pre-vascularized tissue constructs [177]. In vivo, electrospun vascular grafts and patches modified with heparin and VEGF have shown enhanced patency and endothelialization, suggesting better integration into host vasculature [185]. Moreover, heparin coatings reduce the thrombogenicity of implanted materials. For instance, heparin-coated stents have been shown to prevent clot formation while presenting immobilized VEGF to promote re-endothelialization [182]. In a related study, Shen et al. developed a heparin-based coacervate system for FGF2 delivery in myocardial infarction therapy. This formulation protected FGF2 from enzymatic degradation and promoted localized, sustained angiogenesis in infarcted cardiac tissue [186]. Together, these studies underscore how heparin-based platforms facilitate targeted vascular regeneration and mitigate thrombotic risk, key features for both cardiac repair and engineered vascular conduits.

#### 4.2.5. Wound Healing and Chronic Ulcers

Heparin-based systems have shown notable promise in treating chronic wounds and ulcers by enhancing the local availability and stability of pro-healing GFs such as VEGF, FGF, and TGF-β. Topically applied heparin-containing dressings and hydrogels can sequester endogenous GFs, promoting granulation tissue formation and re-epithelialization. For example, platelet-rich fibrin matrices loaded with VEGF, FGF, and PDGF demonstrated significantly prolonged GF retention and enhanced healing outcomes when functionalized with heparin [175]. Heparin has also been found to stimulate GF expression intrinsically. Gansevoort et al. reported that heparin application in diabetic ulcer models upregulated bFGF and TGF-β1 levels, accelerating wound closure and reducing scar tissue [180]. Furthermore, heparin-functionalized scaffolds combining VEGF or FGF2 were shown to substantially improve angiogenesis and dermal regeneration in full-thickness wound models [177]. A coacervate system developed by Wu et al. using heparin and PEAD for sustained FGF2 delivery further demonstrated enhanced re-epithelialization, collagen deposition, and neovascularization, underscoring the role of heparin-based carriers in creating a pro-regenerative wound microenvironment [187]. Overall, heparin-based carriers contribute to wound healing by maintaining a sustained, pro-regenerative microenvironment that supports vascularization and fibroblast activity.

#### 4.2.6. Anti-Fibrotic and Anti-Inflammatory Therapies

Heparin and low-molecular-weight heparins (LMWH) have demonstrated significant anti-fibrotic and anti-inflammatory activity, partly through their ability to modulate GF signaling. Saito et al. reported that LMWH encapsulated in PLGA microparticles effectively reduced lung fibrosis in a bleomycin-induced mouse model, attributed to upregulated hepatocyte growth factor (HGF) expression—a cytokine known for its tissue-protective and fibrosis-inhibitory roles. In vitro studies further supported this mechanism: LMWH-treated lung fibroblasts significantly increased HGF secretion and exhibited reduced fibrotic activation [188].

#### 4.2.7. Cancer Therapy and TME Modulation

Heparin-based nanomaterials are increasingly explored in cancer therapy, where their dual functions—GF sequestration and drug delivery—are leveraged to modulate the TME. Due to its high affinity for angiogenic factors such as FGF2 and VEGF, heparin can inhibit tumor angiogenesis by preventing these GFs from engaging their receptors, thereby suppressing vascularization and tumor growth [188]. LMWH has also demonstrated anti-metastatic properties by disrupting tumor cell adhesion and interfering with chemokine–receptor signaling pathways [182]. In addition, Saito et al. showed that heparin delivery can promote tumor stroma remodeling by upregulating HGF, contributing to the resolution of fibrotic stroma and potentially enhancing therapeutic access.

## 5. Future Perspectives and Research Directions

While heparin-based GF delivery platforms have shown great promise, key challenges remain. One major limitation is the lack of precise spatiotemporal control over multi-GF release, which is essential for processes like angiogenesis and neurogenesis [189]. Current systems often struggle to replicate such complex signaling sequences [190]. Future designs may focus on stimuli-responsive or programmable systems that adapt to physiological cues [18]. Long-term in vivo stability is another concern. Although heparin enhances GF retention, its enzymatic degradation [127] and batch variability can hinder reproducibility [191]. Solutions may include site-specific crosslinking, protective encapsulation, or blending with stable polymers. Emerging tools like machine learning offer new avenues to optimize release profiles based on material properties and biological conditions [192]. Additionally, heparin mimetics may address cost and heterogeneity issues while offering tunable bioactivity. Beyond technical challenges, regulatory hurdles also impede clinical translation. These include the complexity of characterizing heterogeneous heparin conjugates, difficulties in standardizing large-scale manufacturing, and ensuring consistent safety and efficacy across batches. Moreover, regulatory bodies may require rigorous assessment of immunogenicity, long-term biocompatibility, and clearance pathways, especially for chemically modified or crosslinked heparin systems. Advanced fabrication technologies, such as 3D bioprinting, could further enhance delivery precision by incorporating heparin into spatially defined scaffolds [193]. Coupling these systems with gene therapy—for example, co-delivering GF-encoding plasmids—may also enable sustained endogenous expression and improve regenerative outcomes [194]. In summary, advancing heparin-based platforms will require a multidisciplinary effort integrating smart materials, predictive modeling, and regenerative strategies to achieve next-generation therapeutic delivery systems.

## 6. Conclusions

Heparin-based delivery platforms have emerged as powerful and versatile tools for controlled GF release across diverse biomedical applications, particularly in tissue engineering and regenerative medicine. Rooted in heparin’s unique biosynthetic pathway—featuring Golgi-mediated sulfation and the formation of highly anionic glycosaminoglycan chains—these materials exhibit rich structural diversity in sulfation patterns, molecular weight, and chain flexibility. Such characteristics impart strong and selective binding affinities for a broad spectrum of GFs, while also enabling precise control over degradation and release kinetics. As a result, heparin functions not only as a passive carrier but also as a biofunctional regulator of cell signaling and tissue remodeling.

The diversity of heparin-based platforms—spanning injectable hydrogels, self-assembled nanoparticles, electrospun fibers, and coacervates—enhances their adaptability to various clinical needs. Each format offers unique advantages: hydrogels provide in situ gelling and 3D structural support; nanoparticles allow for systemic delivery and intracellular targeting; fibrous scaffolds mimic ECM structures for localized release and cell guidance. The modularity of these systems enables the integration of stimuli-responsive mechanisms—such as pH, temperature, enzymatic activity, redox conditions, and ultrasound—allowing for spatiotemporal control over therapeutic release in response to pathological microenvironments.

Across applications including soft tissue regeneration, bone and cartilage repair, neural and cardiovascular recovery, anti-fibrotic interventions, wound healing, and cancer therapy, heparin-based delivery systems have consistently demonstrated enhanced therapeutic efficacy and biological integration. Looking forward, coupling structural insights from heparin biosynthesis with advanced materials design will enable the development of multi-stimuli-responsive, tissue-specific, and bioadaptive platforms. These innovations are expected to define the next generation of precision medicine tools for regenerative and disease-modulating therapies.

## Figures and Tables

**Figure 1 pharmaceutics-17-01145-f001:**
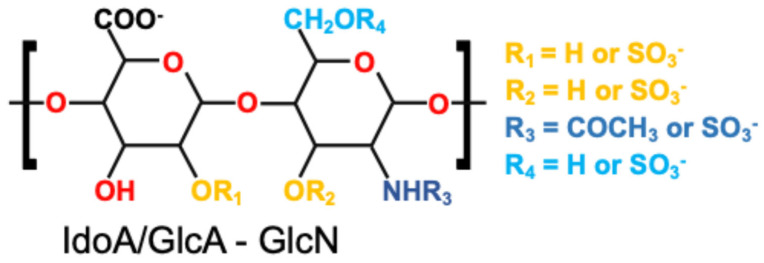
Chemical structure of a representative disaccharide unit of heparin.

**Figure 2 pharmaceutics-17-01145-f002:**
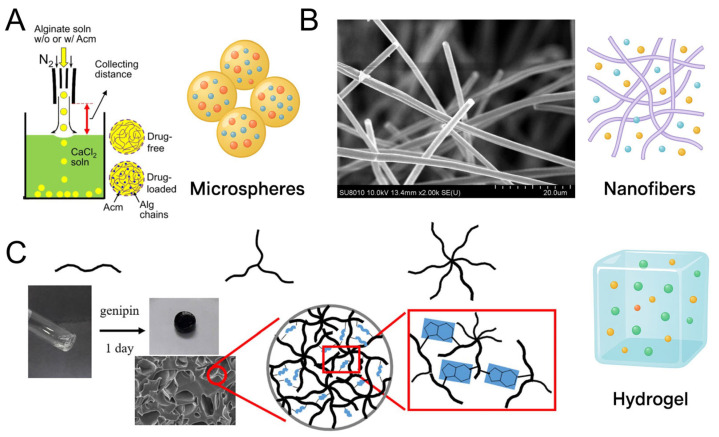
Representative designs of heparin-based drug delivery platforms. (**A**) Schematic and illustration of alginate microspheres formed by ionic gelation. Adapted with permission from [86]. Copyright (2021) Elsevier. (**B**) SEM image and schematic representation of electrospun nanofibers. Adapted from [87]; licensed under CC BY 4.0. (**C**) Crosslinked hydrogel network with embedded drug molecules and porous microstructure. Adapted with permission from [88]. Copyright (2020) Elsevier.

**Figure 3 pharmaceutics-17-01145-f003:**
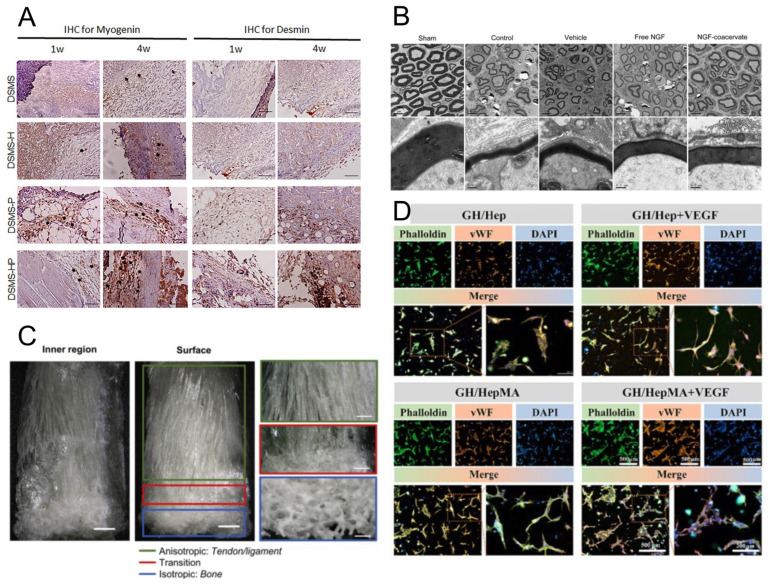
Heparin-based delivery platforms in tissue regeneration: (**A**) Soft tissue. Reproduced from [175], licensed under CC BY 4.0. (**B**) Neural. Adapted with permission from [176]. Copyright (2017) Elsevier. (**C**) Bone. Adapted with permission from [93]. Copyright (2018) Elsevier. (**D**) Vascular applications. Adapted with permission from [177]. Copyright (2024) Elsevier.

**Table 1 pharmaceutics-17-01145-t001:** Functional groups of heparin: chemical locations and corresponding modification strategies.

Functional Group	Chemical Location	Modifiability	Corresponding Design Strategy
–OSO_3_^−^	GlcN, IdoA	Involved in electrostatic and ionic interactions	Electrostatic adsorption, ionic crosslinking
–COOH	GlcA, IdoA	Can be activated for amide formation	Covalent grafting, chemical crosslinking
–OH	GlcN, GlcA, IdoA	Can undergo oxidation, esterification	Covalent grafting, network crosslinking
–NH–COCH_3_	GlcN	Permits enzymatic or chemical conjugation	Covalent grafting, enzymatic crosslinking

**Table 2 pharmaceutics-17-01145-t002:** Heparin loading approaches in delivery platforms.

Strategy	Schematic	Advantages	Challenges	References
Electrostatic Adsorption	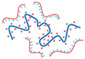	Simple; preserves bioactivity; reversible and suitable for short-term or in situ delivery	Weak interactions; serum- and ion-sensitive;risk of burst release	[33,34,47]
Physical Blending	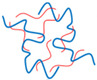	Easy to implement; maintains biomolecule integrity; no harsh reactions	Risk of leaching under physiological conditions;uncontrolled release	[48,49,50]
Covalent Conjugation	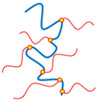	High stability; long-term retention; site-specific functionalization	Requires reaction control; risk of reduced bioactivity; complex chemistry	[51,52,53]
Network Crosslinking	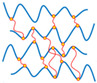	Forms stable 3D networks; enhances mechanical strength; controls release kinetics	Requires reactants; balance mildness and strength	[54,55,56,57,58]

**Table 3 pharmaceutics-17-01145-t003:** Summary of fabrication strategies for heparin-loaded microspheres.

Fabrication	Schematic	Advantages	Challenges	References
Emulsion–Solvent Evaporation/ Extraction	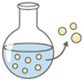	Excellent size tunability Widely used	Use of organic solvents Possible protein denaturation	[89,94,95]
Spray Drying	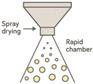	Scalable Rapid productionSuitable for temperature-stable drugs	Not fit for heat-sensitive proteins Limited encapsulation efficiency	[96,97]
Microfluidics	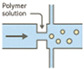	Precise size control High monodispersity Advanced release design	Equipment complexity Low throughput	[98,99,100]
Coacervation	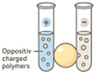	Mild processing Suitable for proteinsElectrostatic compatibility with heparin	Stability of coacervatesBatch-to-batch variability	[75,101,102]
Electrospraying	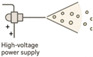	Suitable for sensitive molecules Can form core–shell structures	Solvent useOptimization complexity	[103,104,105]

**Table 4 pharmaceutics-17-01145-t004:** Summary of electrospinning strategies for heparin-loaded fibrous systems.

Fabrication	Schematic	Advantages	Challenges	References
Uniaxial Electrospinning	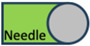	Simple setup, widely used, good fiber uniformity	Limited drug loading complexity, fast release profiles	[128,129]
Coaxial Electrospinning	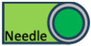	Precise spatial control, reduced burst release, co-encapsulation possible	Requires precise control of flow and viscosity, low scalability	[130,131,132]
Microfluidic-Assisted Electrospinning	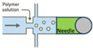	High precision,allows modular design, compatible with in-line crosslinking	Device complexity, clogging, requires stable flow control	[133,134]
Emulsion Electrospinning	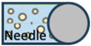	Simple setup, protects sensitive agents, enables multi-phase release	Polydispersity, irregular structure, lower mechanical strength	[135,136]
Triaxial Electrospinning	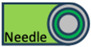	Fine-tuned gradient release, high encapsulation efficiency, structural stability	Complex fabrication, interfacial instability, low throughput	[137,138]
Janus Electrospinning	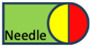	Combines incompatible drugs, biphasic release, targeted surface modification	Jet instability, matching viscosity is critical	[139]
Multilayer Concentric Electrospinning	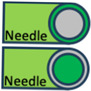	Highly tailored release, Strong protection for proteins, Improved control	High technical demand, Interfacial stability, Poor scalability	[140]

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
