# Peer review of "Heparin-Based Growth Factor Delivery Platforms: A Review"

_pharmaceutics, 2025, doi:10.3390/pharmaceutics17091145_

Round 1
Reviewer 1 Report
Comments and Suggestions for Authors
The review article "Heparin-Based Growth Factor Delivery Platforms: A Review" provides a detailed overview of heparin-based delivery systems for growth factors (GFs) that have been applied in regenerative medicine. The article is organized well, covering the molecular properties of heparin, possible strategies for modifications, potential delivery platforms, and their use. Still, there are some gaps that could be addressed to improve scientific rigor, clarity, and translational relevance of the review. Below are comments and suggestions for improvement:
- The paper is rich with technical details. Because it was intended for a scientific audience, this is fairly standard for the expected reader and might be outside the scope of an interdisciplinary audience, such as clinicians or material scientists who are just beginning to engage with heparin. Consider a brief glossary of key terms (e.g., "electrostatic adsorption," "coacervation") or a simplified introductory version that provides the reader sufficient context to understand heparin for an unclear audience.
-The abstract mentions stimuli-responsive systems (i.e., pH-, enzyme-, or ultrasound-sensitive design), but this topic is not explored numerically in the main text. I would devote a subsection to these systems and their principle, and advantages, and offer examples as GF delivery platforms.
- The introduction should briefly explain why only heparin is suitable for GF delivery compared to other glycosaminoglycans (e.g. chondroitin sulfate or hyaluronic acid).
- While the article provides a well rounded overview of numerous platforms and approaches, it does not evaluate their unique advantages and disadvantages. For instance, a table or discussion evaluating microspheres, nanofibers, and hydrogels regarding particular clinical applications (i.e., injectability, scalability, or stability over time), would strengthen the review.
-The article could benefit from a discussion of trade offs associated with different heparin modification strategies for researchers (e.g., simplicity of electrostatic adsorption vs. the stability of covalent grafting).
-The article could also have a specific area to address current deficiencies in heparin-based delivery systems, such as difficulty in accurately controlling multi-GF delivery, and minimizing loss of in vivo stability in the long-term. Suggest potential future research pathways, such as utilizing machine learning for better optimization of the release kinetics or heparin mimics to alleviate cost concerns. Include discussion of the potential of integrating heparin-based platforms with other technologies, such as 3D bioprinting or gene therapy experiences, to elicit better regenerative outcomes.
- Figures 1-3 have potential, but the legends and annotations could have more descriptive legends/annotations to increase clarity. For instance, in Figure 2, you could label some of the specific interactions (e.g., sulfate groups binding GFs), which will make their mechanisms visually clearer.
-In section 2 (Heparin Structure and Modification Concepts), the authors describe heparin's chemical structure and modification methods in detail using Table 1 and Figure 1 as aids.
The portion of Section 2 dealing with heparin's structure (e.g., disaccharide units, sulfation) could be developed further by describing how different sulfation patterns (e.g., 2-O vs 6-O) modify GF binding affinity and selectivity. This would provide readers with a better understanding of heparin's bioactivity.
Quantitative data could also be provided in Table 1, such as representative binding affinities (e.g., Kd values) for heparin-GF complexes or particular examples of modification yields using specific modification strategies.
- For the sake of clarity, section 3 (Heparin Delivery Platforms) is well organized over three platform types (microspheres, nanofibers, hydrogels) and the way the text categorizes them and describes different types and uses is straightforward enough. The chapter describes fabrication techniques well. You should consider adding a subsection or table that compares the three platforms with respect capsule efficiency, release kinetics, mechanical properties and tolerance of specific comparative tissues, which would help readers determine the best platform for their research or application.
- I recommend adding a summary figure or table that summarizes the fabrication techniques, modification strategies, and applications to serve as a quick guide for readers.
-The references cited are appropriate but some sections, are somewhat based on older studies (i.e. pre-2020). Adding more recent developments (2023-2025) will improve the recency of review, particularly given how fast things are moving in this domain (e.g., biomaterials, drug delivery).
Reviewer 2 Report
Comments and Suggestions for Authors
The manuscript presents a well-structured and timely review. The authors have comprehensively compiled recent developments in the field. I offer the following comments for the authors’ consideration to enhance the quality of the manuscript.
1) In Introduction, clarify comparative advantages of heparin-based systems over other polymeric platforms in terms of release kinetics and clinical translatability.
2) Include more quantitative data (e.g., encapsulation efficiencies, release durations) in the comparative platform tables for better assessment.
3) Mention the limitations of current fabrication techniques like electrospraying and microfluidics, beyond just throughput and scalability.
4) Include mechanistic insights into how heparin’s sulfation patterns specifically affect binding affinity for different growth factors.
5) Discuss regulatory challenges in translating heparin-functionalized delivery systems to clinical applications.
6) Add a graphical abstract.
7) Include representative ex vivo and in vivo results figures to support key findings.
